# Auxin Metabolite Profiling in Isolated and Intact Plant Nuclei

**DOI:** 10.3390/ijms222212369

**Published:** 2021-11-16

**Authors:** Vladimír Skalický, Tereza Vojtková, Aleš Pěnčík, Jan Vrána, Katarzyna Juzoń, Veronika Koláčková, Michaela Sedlářová, Martin F. Kubeš, Ondřej Novák

**Affiliations:** 1Laboratory of Growth Regulators, Institute of Experimental Botany of the Czech Academy of Sciences and Faculty of Science, Palacký University, Šlechtitelů 27, CZ-78371 Olomouc, Czech Republic; vladimir.skalicky@upol.cz (V.S.); vojtkova@ueb.cas.cz (T.V.); ales.pencik@upol.cz (A.P.); 2Institute of Experimental Botany of the Czech Academy of Sciences, Centre of the Region Haná for Biotechnological and Agricultural Research, Šlechtitelů 31, CZ-77900 Olomouc, Czech Republic; jan.vrana@fno.cz (J.V.); kolackova@ueb.cas.cz (V.K.); 3Department of Biotechnology, The Franciszek Górski Institute of Plant Physiology, Polish Academy of Sciences, Niezapominajek 21, 30-239 Krakow, Poland; k.juzon@ifr-pan.edu.pl; 4Department of Botany, Faculty of Science, Palacký University, Šlechtitelů 27, CZ-78371 Olomouc, Czech Republic; michaela.sedlarova@upol.cz; 5School of Life Sciences, University of Warwick, Coventry CV4 7AL, UK

**Keywords:** subcellular fractionation, flow cytometry, nucleus, auxin, auxin metabolism

## Abstract

The plant nucleus plays an irreplaceable role in cellular control and regulation by auxin (indole-3-acetic acid, IAA) mainly because canonical auxin signaling takes place here. Auxin can enter the nucleus from either the endoplasmic reticulum or cytosol. Therefore, new information about the auxin metabolome (auxinome) in the nucleus can illuminate our understanding of subcellular auxin homeostasis. Different methods of nuclear isolation from various plant tissues have been described previously, but information about auxin metabolite levels in nuclei is still fragmented and insufficient. Herein, we tested several published nucleus isolation protocols based on differential centrifugation or flow cytometry. The optimized sorting protocol leading to promising yield, intactness, and purity was then combined with an ultra-sensitive mass spectrometry analysis. Using this approach, we can present the first complex report on the auxinome of isolated nuclei from cell cultures of Arabidopsis and tobacco. Moreover, our results show dynamic changes in auxin homeostasis at the intranuclear level after treatment of protoplasts with free IAA, or indole as a precursor of auxin biosynthesis. Finally, we can conclude that the methodological procedure combining flow cytometry and mass spectrometry offers new horizons for the study of auxin homeostasis at the subcellular level.

## 1. Introduction

The processes of plant growth, development, growth, and plasticity are driven mainly by plant hormones. Auxin, one of the plant hormone groups and represented by native indole-3-acetic acid (IAA), plays an irreplaceable role in all these aspects throughout the entire plant life span. Auxins act at the organ/tissue [1], cellular [2,3], and subcellular level [4]. Thus, precise control of the spatio-temporal distribution of IAA in both plant and cell is critical [5,6].

Auxin homeostasis is tightly regulated by the coordination of transport, biosynthesis, and metabolism, which altogether regulate the availability of IAA. It seems that cellular and subcellular compartmentalization of auxin may be also functionally important for the control of physiological processes [4]. The canonical auxin signaling pathway occurs in nuclei, where IAA “glues” TRANSPORT INHIBITOR RESPONSE 1/AUXIN SIGNALING F-BOX (TIR1/AFB) proteins with AUXIN/INDOLE-3-ACETIC ACID (Aux/IAAs) in a co-receptor complex. Subsequently, Aux/IAAs, which repress AUXIN RESPONSE FACTORs (ARFs), are degraded and auxin-responsive genes are transcribed. Recently, it was shown that TIR1/AFBs and ARF3, also known as ETTIN, are not found exclusively in the nucleus but also in the cytosol. Thus, they may also be involved in non-transcriptional responses to auxin input known as non-canonical auxin signaling [7,8,9,10,11].

Several intracellular auxin transporters have been already described, such as the endoplasmic reticulum (ER) localized PIN-FORMED PIN5, [12] and PIN8, [13,14], PIN-LIKES (PILS) [6,15] and tonoplast located WALLS ARE THIN1 (WAT1) [16]. However, the details of the regulation of intracellular IAA fluxes and thus of the control of subcellular homeostasis remain elusive. Recently, two possible routes—via the ER or via the cytosol—were proposed. IAA can enter the nucleus directly, although ER-to-nucleus flux dominates [17].

De novo IAA biosynthesis is mainly mediated via the TRYPTOPHAN AMINOTRANSFERASE OF ARABIDOPSIS/YUCCA (TAA/YUC) pathway including tryptophan (Trp) and indole-3-pyruvic acid (IPyA) as precursors in *Arabidopsis thaliana* [18]. The TAA/YUC enzyme complex has been shown to be anchored to the ER membrane and faced to the cytosol [19]. However, the YUC enzyme family comprises homologs for which localization remains unclear. Additionally, three other IAA biosynthetic pathways have been described in Arabidopsis named according to their intermediates: indole-3-acetamide (IAM), tryptamine (TRA), and indole-3-acetaldoxime (IAOx) pathway [20]. A Trp-independent pathway for auxin synthesis has also been proposed [21].

Homeostasis of free IAA is likewise controlled by (ir)reversible conjugation with glucose (glc) or amino acids by UDP-glucosyl transferases or GRETCHEN HAGEN 3 (GH3), respectively [22,23,24]. Localization of IAA conjugation is not fully clear, however, it was recently published that GH3.17 occurs in the cytosol [25]. IAA-glucose (IAA-glc) is believed to be a storage form enriched in vacuoles [16], whereas only some IAA amino conjugates can be reversibly transformed back to free IAA by amidohydrolases located mainly in ER [26,27]. DIOXYGENASE FOR AUXIN OXIDATION 1 (DAO1) catalyzes terminal IAA degradation producing 2-oxindole-3-acetic acid (oxIAA) in the cytosol [28,29] which can be also conjugated with amino acids or glucose (oxIAA-glc) [21].

Direct or indirect monitoring of the subcellular auxinome in plants has been a high interest of plant biologists for many decades [30]. An important prerequisite is an effective and robust method providing a pure, intact organelle fraction. A classic biochemical method is differential centrifugation (DC), typically in combination with continuous or discontinuous density gradients. Nuclei are usually pelleted during an initial step at about 1000–2000× *g* as a crude nuclear fraction. For purer fractions, centrifugation using sucrose, Percoll, Ficoll, or other media offers density, osmolarity, or viscosity gradients [31,32,33]. The detergent Triton X-100 was used successfully for endomembrane washing [32,34,35,36].

Two other available methods circumvent typical problems associated with biochemical purification techniques. Flow cytometry (FCM) enables the isolation of nuclei from crude cell lysates [37,38,39]. Fluorescence from a wide range of organelle-specific fluorescent dyes is used, e.g., 4′,6-diamidino-2-phenylindole (DAPI) for nuclei. A method called INTACT (Isolation of Nuclei TAgged in specific Cell Types) allows affinity-based isolation of nuclei. This technique combines specific labeling of a nuclear envelope protein by biotin in the cell type of interest and streptavidin-coated magnetic beads (micro or nano size) for biotin-labeled nucleus affinity purification (AP) [40,41].

In this work, we focus on two subcellular fractionation methods. We optimized methods with emphasis on nuclear intactness, purity, and yield using fluorescence microscopy, immunoblots, and 3D volume reconstruction. Finally, the auxinome of isolated nuclei from cell cultures of Arabidopsis and tobacco was determined by ultra-sensitive LC-MS/MS.

## 2. Results

### 2.1. Comparison of DC and FCM as Nucleus Isolation Methods

Two cell cultures, *Arabidopsis thaliana* ecotype Landsberg *erecta* (*Ath*-L*er*) and tobacco *Nicotiana tabacum* cv. Bright Yellow 2 (BY-2) were cultivated under the same standard conditions (see details in the Methods section). The cell lines showed distinct characteristics, such as the formation of cell clusters (Figure 1a,f), different growth parameters such as cell density and growth curves [42], and composition of the cell wall [43]. Therefore, both lines were selected for our comparative study.

We tested various protocols for releasing nuclei from the plant cells. The first isolation protocol was based on grinding cell cultures using a dounce homogenizer in combination with classical DC [44] (Method S1). Cell debris was present in the nuclear fraction (Appendix A) and so we decided to examine homogenization protocols based on protoplast isolation [34,45] (Figure 1b,g). A protoplasting protocol based on Saxena et al. [34] was optimized, although problems with quality and yields persisted. Therefore, we finally applied the well-established protocol published by Yoo et al. [45]. Importantly, plasma membrane rupture enabled the soft release of intact subcellular compartments during the protoplast lysis step (Figure 1c,h). Samples were further processed to obtain purified, enriched intact nuclei (Figure 1c,e,h,j).

The purity of all nuclear fractions was evaluated by immunoblot analysis (Figure 2). Four representative organelle markers for nuclei (Histone 3, H3), ER (Lumena-binding protein, BiP), Golgi complex (Coatomer subunit gamma, Sec21p), and vacuole (Epsilon subunit of tonoplast H^+^-ATPase, V-ATPase) were selected. Compared to the nuclear fractions isolated by the FCM method (Figure 2b), fractions isolated by the DC method showed the presence of contaminating organelle markers, mainly ER and Golgi complex (Figure 2a). Next, we performed volume analysis of sorted nuclei together with image processing and 3D-FISH according to Koláčková et al. [46] and Perničková et al. [47] (Method S2). Our data clearly showed that cell nuclei isolated by FCM offer intactness, but also heterogeneity of 3D shapes. However, the calculated volumes based on 3D reconstructions suggest a low level of variability (Appendix A). In summary, FCM provides lower yields, but significantly higher purity of intact nuclei than the classical DC method. Moreover, the use of FCM can improve subsequent LC-MS/MS analysis [48].

### 2.2. Auxinome of Isolated Nuclei

To reduce the time required for the preparation of individual samples and to improve nuclear intactness, a formaldehyde fixation-based method [49] (Method S1) was tested. Surprisingly, fixation was not compatible with auxin metabolic analysis. Formaldehyde cell fixation significantly reduced sensitivity and/or caused a loss of MS signal, including the internal standards.

Therefore, a long-term stability control experiment was performed to evaluate the effect of the FCM procedure on auxin profiles in *Ath*-L*er* nuclei. All sorted fractions were analyzed by LC-MS/MS [50]. Surprisingly, no dramatic conversion between IAA, its main precursor (IPyA), and catabolite (oxIAA) was observed in sorted nuclei up to 14 h after protoplast lysis (Figure 3a). This finding indicates valid results of endogenous auxin levels in nuclear fractions isolated by the FCM method.

A comparison of IAA metabolic profiles in *Ath*-L*er* nuclei isolated by DC or FCM showed that only six auxin metabolites were quantified in the fraction isolated by DC, while the FCM approach gave a wider portfolio of compounds across the auxinome (Figure 3b). This optimized approach combining an enzymatic cell wall digestion and FCM coupled with an ultra-sensitive MS-based analysis was further used to profile the auxinome in isolated nuclei from both *Ath*-L*er* and BY-2 cell cultures. The analysis reveals not only IAA and relative metabolites, but also its precursors to give the first complex auxin metabolic profile for plant nuclei (Figure 1). Nine auxin-related compounds were detected, although IAM was detected only in *Ath*-L*er* nuclei (Appendix A). Surprisingly, indole-3-acetonitrile (IAN) was not detected in the isolated nuclei compared to the analysis performed on *Ath*-L*er* cells (Appendix A). Moreover, IAA-glc and oxIAA-glc were not detected either in nuclei or cells.

The dominant compound in all samples was tryptophan, 99.6% and 99.8% of relative distribution in *Ath*-L*er* and BY-2, respectively (Figure 4a). Interestingly, the BY-2 results showed a higher proportion of IAA precursors (IPyA; TRA; and anthranilate, ANT), whereas *Ath*-L*er* profiles had significantly higher levels of IAA and its metabolites (oxIAA; IAA-aspartate, IAAsp; and IAA-glutamate, IAGlu) (Figure 4a and Appendix A). ANT and IAGlu not detected in whole *Ath*-L*er* cells (Appendix A) were detectable in isolated nuclei (Appendix A). In conclusion, our findings illustrate that we are able to detect credible nuclear auxin profiles from two plant cell models using an optimized isolation protocol based on the FCM technique and disclose metabolites that are under detection limit in whole-cell fraction.

### 2.3. Feeding Experiments with Indole and IAA

Finally, to demonstrate the reliability of the FCM-based isolation method, we decided to promote auxin metabolism in *A. thaliana* protoplasts by treatment (3 h) with 10 µM indole as a precursor of the Trp-dependent biosynthetic pathway [20]. The data were converted to the ratio of auxin levels in treated and untreated nuclei (Figure 4b). Our results showed that the level of indole precursor ANT decreased after indole treatment. Compared to the non-treated control, we clearly detected increased levels of IAA precursors (Trp, TRA, IAM and IPyA), in the range of 1.4- to 3.8-fold. As expected, IAN levels were below the limit of detection. Interestingly, the IAA level was not affected by indole treatment, however, slightly elevated levels of oxIAA and IAAsp were detected (Figure 4b). Due to the negligible increase in IAA and its metabolites, protoplasts were fed with 100 µM IAA for 1 h. Under our experimental conditions, levels of not only free IAA but also other analyzed metabolites such as oxIAA, IAAsp, and IAGlu were very significantly increased (Figure 4b). However, the levels of all IAA precursors were not affected by IAA treatment except for a slight increase in IPyA level (Appendix A). Confirming our previous results, the glucose conjugates IAA-glc and oxIAA-glc were not detected.

Overall, our results of the feeding experiment nicely showed the applicability of the organelle-isolation method based on FCM. In the near future, we should be able to quantify not only auxins but also other plant hormones in sorted organelle populations.

## 3. Discussion

The maintenance of auxin homeostasis and its distribution within the plant cell remains elusive because only a few previous studies were focused on subcellular auxin analysis in chloroplasts [51], vacuoles [16], and recently ER [52], but none in nuclei. The nucleus represents a key organelle because it contains the process of canonical auxin signaling. Therefore, we decided to examine several protocols for nuclei isolation with subsequent auxin analysis by LC-MS/MS.

It is important to emphasize that the nuclei isolation protocols taken as the basis for this study were designed mainly for DNA or genomic studies [35,53]. Nevertheless, the intactness and high purity of the isolated nuclei are crucial for auxin metabolic profiling. Release of nuclei from plant tissue or cell cultures can be achieved by various homogenization methods but a combination of enzymatic cell wall digestion and osmotic plasma membrane disruption was found to be most conducive for gentle release of nuclei. This protoplast preparation has been successfully employed for organelle isolation in previously published protocols [34,54,55] and the protoplast lysis step does not significantly change auxin metabolism [56] which remains stable long-term after nucleus isolation as our data show (Figure 3a). Immunoblot analysis and confocal microscopy verified the success of the protocols used (Figure 1 and Figure 2). In general, DC showed low resolving power leading to the presence of contaminating organelles in the isolated nuclear fraction [33]. To reduce contaminants, Triton X-100 is often used as a non-ionic detergent in DC protocols [57]. However, as seen in Figure 2, neither the addition of a small amount of detergent did not lead to higher purity compared to the FCM method. In conclusion, the FCM approach provides better purity compared to DC-based nuclei isolation (Figure 2), and 3D nuclei reconstruction showed that FCM nuclei were intact and with only minor shape heterogeneity (Appendix A).

Quantitative analysis of auxins at the cellular or even subcellular level is challenging [58] although recent improvements of plant sample purification protocols and mass spectrometry-based techniques have enabled the analysis of the auxinome in minute samples (~2 mg of fresh weight; [59]). The use of additional steps, such as organelle isolation, should further reduce the contents of pigments, lipids, phenolic compounds, and other interfering compounds in analyzed samples. Overall, we observed a better signal during LC-MS/MS analysis in samples sorted by FCM compared to DC-delivered nuclei, indicating a reduced matrix effect (Figure 3b).

Profiling of the auxinome in nuclear fluid revealed the presence not only of free IAA but also the precursors of three biosynthetic pathways (IPyA, IAM, and TRA; [4,20]) as well as IAA main catabolite oxIAA and conjugates with Asp and Glu (Figure 4a,b). Surprisingly, no ester-linked sugar conjugates IAA-glc and oxIAA-glc were detected in isolated nuclear fractions, even after the feeding with IAA. This, together with the fact that IAA-glc and oxIAA-glc have not been determined in whole *Ath*-L*er* cells (Appendix A), suggests that the *Ath*-L*er* cell culture has a reduced ability to form ester-linked conjugates of IAA and oxIAA. How and why IAA precursors and metabolites are transported into the nucleus is still unclear, although Middleton et al. [17] showed that ER-to-nucleus auxin flux represents a major subcellular pathway controlling nuclear auxin levels, and that auxin diffusion from the cytosol via the nuclear pores plays only a minor role. Additionally, it was shown that PILSes can reduce nuclear auxin signaling in the apical hook leading to the de-repression of growth and the onset of the hook through the direct regulation of PILS gene activity by phytochrome B-reliant light-signaling pathway [60].

We should also mention the roles of other organelles in auxin homeostasis. For example, Ranocha et al. [16] detected IAA, its precursors and metabolites in vacuoles, and Včelařová et al. [52] recently in ER. Nevertheless, the biological importance and function of compartmentation in the nucleus and other organelles remains elusive and needs to be further studied.

## 4. Materials and Methods

### 4.1. Cultivation of Cell Lines

Arabidopsis (*Arabidopsis thaliana* cv. Landsberg *erecta*) cell line (*Ath*-L*er*) [61] was grown in sterile Murashige-Skoog medium (4.4 g·L^−1^, pH 5.8; Duchefa Biochemie, Haarlem, The Netherlands) supplemented with vitamins, 3% sucrose, 0.232 µM kinetin and 5.37 µM 1-naphthaleneacetic acid (Merck Life Science, Darmstadt, Germany). Tobacco (*Nicotiana tabacum* cv. Bright Yellow 2) cell line (BY-2) [62] was grown in sterile Murashige-Skoog medium (4.3 g·L^−1^ pH 5.8) supplemented with 3% sucrose, 4 µM thiamin, 555 µM inositol, 1.47 mM KH_2_PO_4_, and 0.9 µM 2,4-dichlorophenoxyacetic acid (Merck Life Science, Darmstadt, Germany). Both cell lines were subcultured weekly into fresh media in a volume ratio of 1:10. The cells were cultivated at 23 °C in the dark and shaken at 120 rpm. Five-day-old cells were used for all experiments. As a stock material, cell calli of both cell lines were cultivated on the same solidified media (1.2% agar) and subcultured monthly.

### 4.2. Protoplast Preparation

The cells were gently filtered (0.45 µm nylon filter) using vacuum filtration. Protoplasts were prepared according to Yoo et al. [45] with minor modifications in enzyme combination and final concentrations. Exactly 6 g of cell material was resuspended in 60 mL of preheated protoplasting buffer (37 °C; 0.6 M mannitol, 10 mM KCl, 2 mM 4-morpholineethanesulfonic acid (MES), 2 mM CaCl_2_, 0.1% (*v*/*v*) BSA, 2 mM MgCl_2_, 20 U/mL cellulase Onozuka R-10, 7.5 U/mL macerozyme R-10, 0.3 U/mL pectolyase Y-23, and 45 U/mL celulysin pH 5.7; Merck Life Science, Darmstadt, Germany, enzymes were from Duchefa Biochemie, Haarlem, The Netherlands). The suspension was shortly incubated at 37 °C to activate the enzymes and then incubated at 28 °C for 3 h with gentle and occasional hand-shaking. The enzyme solution with released protoplasts was then diluted with an equal volume of preheated W5 buffer (2 mM MES, 154 mM NaCl, 149 mM CaCl_2_ and 5 mM KCl, pH 5.7, 28 °C) to stop enzymatic reaction. The protoplast suspension was gently filtered through a pre-wetted nylon mesh (70 µm) to remove undigested cell clusters and the released protoplasts were pelleted by centrifugation (100× *g*, 3 min, 20 °C). The pelleted protoplasts were gently washed twice in 2–3 mL of WI buffer (4 mM MES, 0.5 M mannitol, 20 mM KCl and 1% sucrose, pH 5.7) by repeating both centrifugation and resuspension steps. We checked the intactness and viability of isolated protoplasts using a microscope and a counting chamber. Importantly, any sample of isolated protoplasts that showed a loss of integrity and intactness higher than 5% was omitted for further processing. The released protoplasts were finally stored on ice. A cell culture protoplasting protocol based on the protoplast lysis step published by Saxena et al. [34] was also tested.

For feeding experiments, the protoplast culture was incubated with 10 µM indole (final concentration) for 3 h (compound added at the beginning of protoplast isolation protocol) or 100 µM IAA (final concentration) for the last 1 h of protoplasting process.

### 4.3. Nuclei Isolation by Differential Centrifugation

The protoplast culture was prepared based on a modified protocol from Yoo et al. [45] (for details see Section 4.2) and subsequent nuclei isolation by DC was performed according to Saxena et al. [34] with minor modifications. All isolation steps were carried out at 4 °C and with pre-cooled buffers. Briefly, the protoplast pellet was resuspended in 1.5 mL of 0.6 M mannitol and 7.5 mL of 20% (*w*/*v*) sucrose, which was added slowly to avoid mixing of the layers. The next step was centrifugation (100× *g*, 7 min, 4 °C) to effectively separate the protoplasts into the mannitol phase. The collected ring of protoplasts was mixed with 13 mL of nuclear isolation buffer (NIB; 200 mM sucrose, 10 mM MES, 10 mM NaCl, 10 mM KCl, 0.1% Triton X-100, 2.5 mM dithiotreitol and 0.1 mM spermine, pH 5.3; Merck Life Science, Darmstadt, Germany) and then incubated on ice for 15 min. To release a satisfactory number of nuclei, the suspension was mixed for 1 min on a shaker, filtered through the pre-wetted three layers of the Miracloth membrane (20–25 µm) and then pelleted by centrifugation (150× *g*, 8 min, 4 °C). The obtained nuclear fraction was resuspended in 2 mL of NIB buffer without Triton X100 and centrifuged (150× *g*, 8 min, 4 °C). The final pellets of isolated nuclei were immediately used for microscopy analysis or frozen in liquid nitrogen and stored at −80 °C.

### 4.4. Nuclei Isolation by FCM

The nuclei were isolated from the protoplasts prepared as described in Section 4.2 using the FCM method according to Petrovská et al. [49] with minor modifications [48]. All samples were mixed with an equal volume of 0.7% NaCl to disrupt the protoplast plasma membranes. The released nuclei were then filtered through 20 µm nylon mesh and stained with DAPI (final concentration 2 µg/mL; Merck Life Science, Darmstadt, Germany). Sorting of nuclei was performed in 0.7% NaCl as a sheath fluid solution using a BD FACS Aria II SORP flow cytometer (BD Bioscience, Franklin Lakes, NJ, USA) equipped with a nozzle (70 µm diameter), filtration bandpass 450/30 nm for DAPI and 480/10 nm for flow cytometry standard (FCS), system pressure 482.6 kPa, UV laser (355 nm, 100 mW), and blue laser (488 nm, 100 mW) for FCS (DAPI-A for sorting of nuclei population, DAPI-W vs. DAPI-A for sorting). The population of nuclei was selected according to the following optical parameters: forward and side scatter in combination with DAPI specific fluorescence. Nuclear fractions were immediately used for fluorescent microscopy and/or frozen in liquid nitrogen.

### 4.5. Microscopy Analysis

The purity of the nuclear fraction and the intactness of the isolated nuclei were verified by a fluorescent microscope Olympus IX51 (Olympus, Tokyo, Japan). Nuclei were stained with DAPI (final concentration 1 µg/mL). The number of nuclei isolated by DC was estimated using a Bürker counting chamber (~8–10 samples were calculated).

### 4.6. Immunoblot Analysis

The purity of nuclear fractions isolated by DC or FCM were also checked by immunoblot analysis utilizing a set of antibodies against protein organelle markers. The pellet of nuclei (approx. 1–3 million) was suspended in Laemmli sample buffer and boiled at 95 °C for 5 min. Nuclear proteins were separated using SDS-PAGE on a 12% polyacrylamide resolving gel and a 4% stacking gel at 90 V for 0.5 h and then at 120 V for ~1.5 h [63] (Bio-Rad, Hercules, CA, USA). The separated proteins were electrophoretically transferred onto a 0.45 μm nitrocellulose membrane at 290 mA for 2 h (Santa Cruz Biotechnology, Heidelberg, Germany). The membrane was blocked in 5% low-fat milk dissolved in TBST buffer for 1 h, incubated for 1 h with primary rabbit antibodies (Agrisera, Vännäs, Sweden): Sec21p (1:1000 diluted; AS08 327), CNX1/2 (1:2500 diluted; AS12 2365), BiP (1:2500 diluted; AS09 481), V-ATPase (1:2000 diluted; AS07 213), H3 (1:5000 diluted; AS10 710). After washing three times with TBST buffer for 5 min, the membrane was incubated with secondary antibody goat anti-rabbit IgG (H&L) conjugated with HRP (1:10,000 diluted; AS09 602) for 1 h. Proteins were visualized by SuperSignal^®^ West Pico Chemiluminiscent substrate (Thermo Scientific, Waltham, MA, USA) according to the manufacturer’s instructions using a ChemiDoc MP Imaging System (Bio-Rad, Hercules, CA, USA).

### 4.7. Auxin Purification and Determination

Full auxinome (IAA, its precursors, and metabolites) was determined as described in Novák et al. [50]. First, the pelleted nuclei isolated by DC approach were suspended in 1 mL sodium-phosphate buffer (pH 7.0). Aliquots of around 1 million nuclei were used for each technical replicate. Next, the sorted nuclei (3 million nuclei for replicate) in sheath fluid (0.7% NaCl) were diluted with deionized water up to 1 mL. Prior auxin extraction, a cocktail of internal standards (50 pmol of [2H5]TRA and [2H4]TRA, 10 pmol of [2H4]IAN and 5 pmol of [2H4]ANT, [2H5]IAM, [2H4]IPyA, [13C6]IAA, [13C6]oxIAA, [13C6]IAAsp, [13C6]IAGlu, [13C6]IAA-glc and [13C6]oxIAA-glc) was added to each sample. The extracts were acidified with HCl to pH 2.7 and then purified by solid-phase extraction (SPE) using OasisTM HLB columns (30 mg/mL, Waters). For quantification of IPyA, separate samples were derivatized by cysteamine (0.25 M, pH 8.0) for 1 h, acidified with HCl to pH 2.7, and purified by SPE. All analytes were eluted with 80% methanol and the eluents were then evaporated to dryness. Auxin profiles were determined by an ultra-high-performance liquid chromatography–electrospray tandem mass spectrometry using an Acquity UPLCTM System (Waters Corp., Milford, CT, USA) equipped with Kinetex C18 (50 mm × 2.1 mm, 1.7 µm; Phenomenex) coupled to a triple-quadrupole mass spectrometer (Xevo TQ-S MS; Waters Corp., Milford, CT, USA) [50].

The long-term stability of auxin profiles in nuclei isolated by the FCM method was tested using two sets of nuclear-isolated fractions. Samples (*Ath*-L*er* cells) were sorted directly or stored on ice (14 h) after protoplast lysis and then analyzed by LC-MS/MS.

## 5. Conclusions

An innovative combination of FCM with ultra-sensitive mass spectrometry analysis has been shown to provide a useful tool for monitoring IAA and other related compounds at the subcellular level. Our methodology has given unprecedented information about the subcellular distributions of the auxinome and thus should facilitate attempts to elucidate regulatory networks involved in plant developmental processes.

Finally, we summarized the main pros and cons of different nuclei isolation approaches, such as conventional biochemical methods of DC or density gradient centrifugation (GC), and modern methods exploiting the technique of FCM or AP. We focused on five main parameters of cell nuclei isolation such as yield, purity, instrumentation, duration, and cell-type specificity (Table 1). We want to point out that these techniques can be easily applied to most cell organelles and the FCM technique could be used to sort more than one or two organelle populations from the same biological material simultaneously. FCM represents a powerful tool for subcellular fractionation with following downstream applications including various “omics” approaches [49,53,64,65].

## Figures and Tables

**Figure 1 ijms-22-12369-f001:**
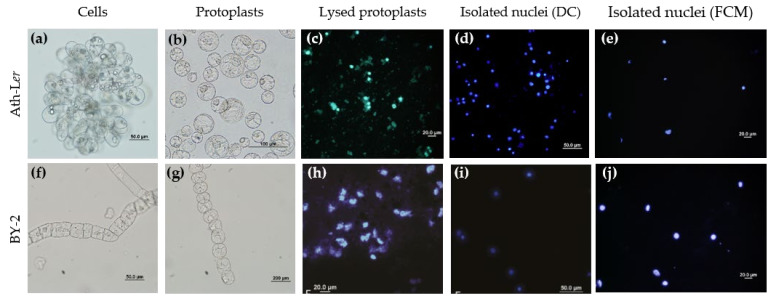
Microscopic analysis of nucleus isolation protocols. Isolations of nuclei from cell lines of (**a**–**e**) *Arabidopsis thaliana* ecotype Landsberg *erecta* (*Ath*-L*er*) and (**f**–**j**) *Nicotiana tabacum* cv. BY-2 (BY-2) were performed by differential centrifugation (DC) or flow cytometry (FCM). (**a**) *Ath*-L*er* cells, (**b**) *Ath*-L*er* protoplasts, (**c**) *Ath*-L*er* protoplasts after lysis, (**d**) *Ath*-L*er* nuclei isolated by DC, (**e**) *Ath*-L*er* nuclei isolated by FCM; (**f**) BY-2 cells, (**g**) BY-2 protoplasts, (**h**) BY-2 protoplasts after lysis, (**i**) BY-2 nuclei isolated by DC, (**j**) BY-2 nuclei isolated by FCM. Nuclei were stained with DAPI (1 µg/mL).

**Figure 2 ijms-22-12369-f002:**
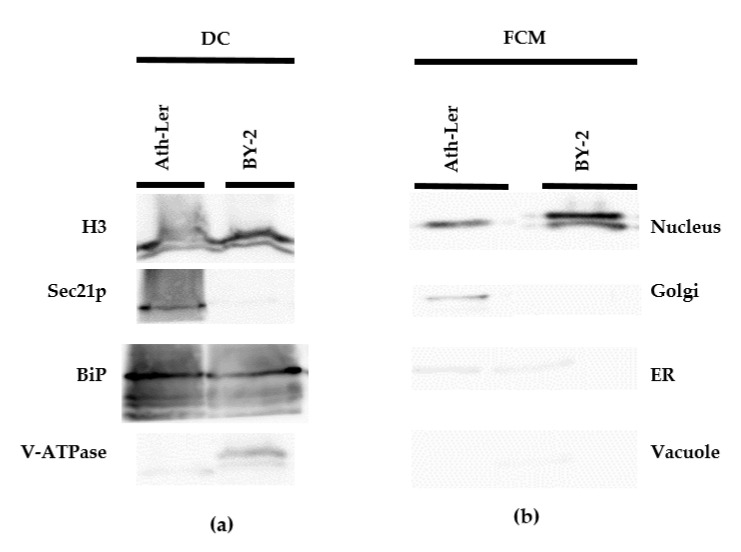
Purity control of nuclear fractions by immunoblotting. Nuclei were isolated from *Arabidopsis thaliana* L*er* (*Ath*-L*er*) and *N. tabacum* cv. BY-2 (BY-2) cell lines by (**a**) differential centrifugation (DC) and (**b**) flow cytometry (FCM). Nuclear extracts were immunoblotted using anti-Histone 3 (H3) to confirm enrichment of nuclei. The following antibodies were used to test for the presence of contaminating organelles: Anti-Coatomer subunit gamma (Sec21p) for presence of Golgi complex, anti-Lumenal-binding protein (BiP) for the presence of endoplasmic reticulum (ER), and anti-Epsilon subunit of tonoplast H^+^-ATPase (V-ATPase) for the presence of vacuole.

**Figure 3 ijms-22-12369-f003:**
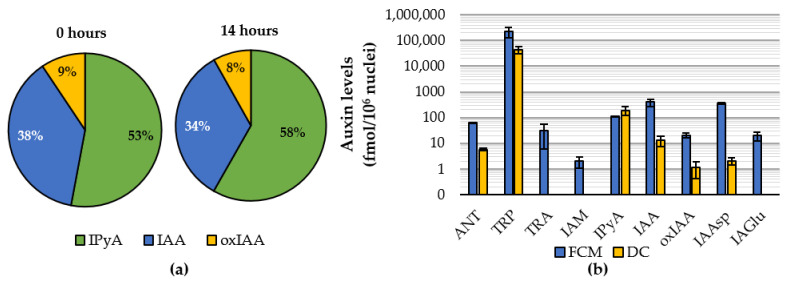
Auxin metabolite profiles in isolated nuclei of *Arabidopsis thaliana-*L*er*. (**a**) Stability of auxin metabolite profile based on relative distribution of indole-3-pyruvic acid (IPyA), indole-3-acetic acid (IAA) and 2-oxoindole-3-acetic acid (oxIAA) in *Ath*-L*er* nuclei sorted directly (0 h) or stored on ice (14 h) after protoplast lysis. (**b**) The auxinome determined in *Ath*-L*er* nuclei obtained by differential centrifugation (DC) or flow cytometry (FCM). Anthranilate (ANT), tryptophan (TRP), tryptamine (TRA), indole-3-acetamide (IAM), IAA-aspartate (IAAsp), IAA-glutamate (IAGlu). Indole-3-acetonitrile, IAA-glucose and oxIAA-glucose were not detected. Five and four biological replicates were analyzed for nuclei isolated by DC and FCM, respectively, except IPyA (*n* = 2/3). Error bars indicate sd.

**Figure 4 ijms-22-12369-f004:**
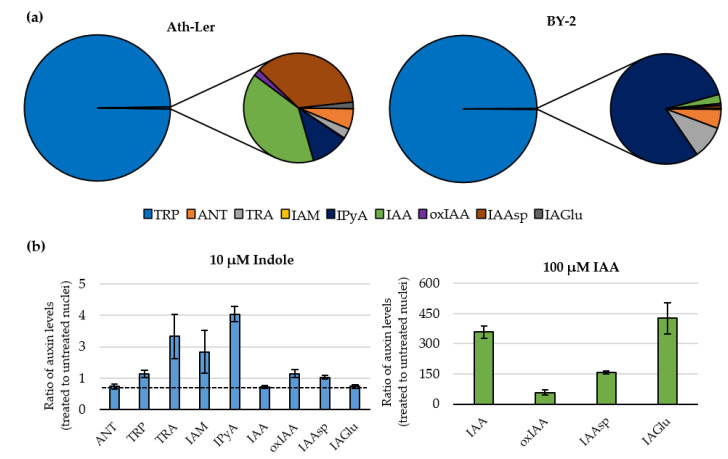
(**a**) Auxinomes in nuclei sorted by flow cytometry from *Arabidopsis thaliana* L*er (Ath*-L*er*; left) and *N. tabacum* cv. BY-2 (BY-2; right), (*n* = 4). (**b**) Relative auxinomes in nuclei after treatment of *Ath*-L*er* protoplasts with 10 µM indole 3 h (left) and 100 µM indole-3-acetic acid 1 h (IAA; right). The auxin concentration was calculated as fmol/1,000,000 nuclei, and the respective ratios (treated to untreated) were then determined (*n* = 5–6). Anthranilate (ANT), tryptophan (TRP), tryptamine (TRA), indole-3-acetamide (IAM), indole-3-pyruvic acid (IPyA), IAA-aspartate (IAAsp), IAA-glutamate (IAGlu), and 2-oxoindole-3-acetic acid (oxIAA). Indole-3-acetonitrile (IAN), IAA-glucose, and oxIAA-glucose were under the limit of detection. Error bars indicate SD.

**Table 1 ijms-22-12369-t001:** Methodology overview of different methods of plant nucleus isolation. The comparison of their key parameters is distinguished accordingly to their (dis)advantages. Differential centrifugation (DC), gradient centrifugation (GC), flow cytometry (FCM), affinity purification (AP), genomics (G), proteomics (P), transcriptomics (T), metabolomics (M).

Parameters	DC	GC	FCM ^1^	AP
Yield	+++	++	+	−
Purity	−	+	++	+++
Instrumentation	+++	+++	−	+
Duration	+	−	++	+++
Cell-type specificity	−	−	+++1	+++
Simultaneous multi-organelle isolation	−	−	+++	−
Downstream application	G, P, (M)	G, T, P	G, P, T, M	G, T

^1^ FCM is also able to sort nuclei into different phases of the cell cycle.

## Data Availability

Data presented in the current study are available in the article and Appendix A.

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
