# Peer review of "Auxin Metabolite Profiling in Isolated and Intact Plant Nuclei"

_ijms, 2021, doi:10.3390/ijms222212369_

Round 1
Reviewer 1 Report
This manuscript describes the comparison of four methods of nuclear isolation from Arabidopsis plants: the yields, extent of contaminating ER and Golgi fractions, and suitability for studying uptake and processing of IAA and IAA metabolites in nuclei.
The work is clearly written, and well-substantiated with clear figures. The work should be a significant reference for all researchers interested in nuclei isolation from plants.
There are a small number of very minor grammatical errors that should be fixed prior to publication. For example:
line 23 - methods of nucleus isolation (nuclear)
line 88 - Other two available methods (The other two, or, Two other available methods...)
Author Response
This manuscript describes the comparison of four methods of nuclear isolation from Arabidopsis plants: the yields, extent of contaminating ER and Golgi fractions, and suitability for studying uptake and processing of IAA and IAA metabolites in nuclei.
The work is clearly written, and well-substantiated with clear figures. The work should be a significant reference for all researchers interested in nuclei isolation from plants.
- We are glad to hear that reviewer 1 was satisfied and asked for some minor text revisions.
There are a small number of very minor grammatical errors that should be fixed prior to publication. For example:
line 23 - methods of nucleus isolation (nuclear)
- The text has been modified as reviewer suggested.
line 88 - Other two available methods (The other two, or, Two other available methods...)
- The text has been edited according to the reviewer's recommendations.
Reviewer 2 Report
The authors of the manuscript “Auxin metabolite profiling in isolated and intact plant nuclei” described protocols for studying auxin homeostasis at the subcellular level. The described methods, such as flow cytometry and mass spectrometry, are easy to follow and allow the detection of IAA and other related compounds at the subcellular level. The metabolites levels may alter by enzymatic cell wall digestion; however, the results show auxin metabolism remains stable long-term after nucleus isolation from protoplasts. The results reported are encouraging. The manuscript is well written. I recommend this paper to be accepted after a minor revision.
Some of the specific comments are as follows:
Introduction: Why did the authors choose protoplasts for the isolation of nuclei? Also, please add information on the advantages and drawbacks of methods used to isolate nuclei and metabolite profiling in this study.
L105-108: Belongs to introduction.
L119: Figure 1 caption: Isolated nuclei (FCM) instead of Isolated nuclei (FC).
L128-130: Please discuss the results [L231-234].
L138: Figure 2 caption: FCM instead of FC.
L190: A. thaliana (Italicize).
L264-267: Please indicate the source of cell lines or provide the establishment methods.
L294: Please indicate the yield and viability of the protoplasts.
Author Response
The authors of the manuscript “Auxin metabolite profiling in isolated and intact plant nuclei” described protocols for studying auxin homeostasis at the subcellular level. The described methods, such as flow cytometry and mass spectrometry, are easy to follow and allow the detection of IAA and other related compounds at the subcellular level. The metabolites levels may alter by enzymatic cell wall digestion; however, the results show auxin metabolism remains stable long-term after nucleus isolation from protoplasts. The results reported are encouraging. The manuscript is well written. I recommend this paper to be accepted after a minor revision.
- We are glad to hear that reviewer 2 was also satisfied with the quality of our manuscript and asked for some minor comments to be reviewed.
Some of the specific comments are as follows:
Introduction: Why did the authors choose protoplasts for the isolation of nuclei? Also, please add information on the advantages and drawbacks of methods used to isolate nuclei and metabolite profiling in this study.
- We paid an attention to integrity of nuclei due to downstream auxin analysis. Conventional homogenization of plant material under liquid nitrogen can cause organelle disruption (Song et al., 2006; doi: 10.1002/pmic.200500893). Other homogenization methods (Dounce homogenizer, Potter-Elvehjem PTFE pestle and glass tube) applied to the cell culture were not enough effective. For this reason, we decided include cell wall digestion.
- A methodological overview of the different methods of plant nucleus isolation, including downstream applications, is summarized in Table 1. The comparison of their key parameters are distinguished accordingly to their (dis)advantages. In context with auxin profiling, the buffers used for differential centrifugation (DC) contains higher concentration of ions interfering with LC‑MS/MS analysis and negatively affect the final determination of analytes (see in Figure 3b). Furthermore, the flow‑cytometric method provide better purity of the nuclear fraction as well as the number of sorted nuclei. Moreover, they are sorted in shorter time than isolated by DC. Using the DC-based approach, isolated nuclei must be counted employing a fluorescence microscope and a counting chamber.
L105-108: Belongs to introduction.
- Thank you for your comment. We would rather keep this short paragraph about the cell lines characteristics and specific features in results section. It is an important information basis of reader to understand why those two standard cell lines were selected for the presented research. Moreover, the distinct characteristics of cell lines are shown as results in Figure 1a,f.
L119: Figure 1 caption: Isolated nuclei (FCM) instead of Isolated nuclei (FC).
- Figure was modified as reviewer suggested.
L128-130: Please discuss the results [L231-234].
- The Discussion section have been slightly modified as suggested by the reviewer. Following sentences have been added: “To reduce contaminants, Triton X-100 is often used as a non-ionic detergent in DC protocols (McKeown et al. 2008; doi:10.1007/978-1-59745-406-3_5). However, as seen in Figure 2, neither the addition of a small amount of detergent did not lead to higher purity compared to the FCM method.”
- The purity of isolated nuclei is usually verified by microscopy or PCR-based methods. Unfortunately, we did not find a proper reference where the purity was tested by immunoblotting.
L138: Figure 2 caption: FCM instead of FC.
- Figure was modified as reviewer suggested.
L190: A. thaliana (Italicize).
- Done.
L264-267: Please indicate the source of cell lines or provide the establishment methods.
- To our knowledge, tobacco BY-2 cells Nicotiana tabacum L. cv. Bright Yellow 2 (Nagata et al., 1992), and Arabidopsis thaliana ecotype Landsberg erecta (May and Leaver, 1993) cell lines were originally established in the early 1990s and have been cultured continuously for the past 30 years. Moreover, both lines are widely used as a standard plant cell line in many applications of plant research besides Arabidopsis thaliana, ecotype Columbia, whose phenotype is more clustery. References to plant cell lines have been added.
L294: Please indicate the yield and viability of the protoplasts.
- Reviewer’s comment is absolutely correct, thank you for noticing us. Detailed information has been added.
- We would like to point out that the main goal was the isolation of intact nuclei and the protoplast isolation was only a necessary inter- step to provide more gentle and standardized conditions for the isolation of nuclei and further nuclei processing. The main goal of our protoplast isolation protocol was to maximize the yield (not exactly comparable to the classical application of protoplasts in microscopy, molecular and cell biology approaches) for the next steps that lead to the isolation of intact nuclei. Due to the limited time and continuity of the further protocol steps, we focused mainly on the integrity of protoplasts. We checked their intactness and viability using a microscope and a counting chamber. Simply saying, any sample of isolated protoplasts that showed a loss of integrity and intactness higher than 5% was omitted for further processing. Importantly, this was quite rare and the isolation protocol applied showed sufficient robustness.